# Effects of Tai Chi on anxiety and theta oscillation power in college students during the COVID-19 pandemic: A randomized controlled trial

Min Wang[1,2]*, Shuxun Chi[1], Xingze Wang[1], Tongling Wang[1]

1 Institute of Physical Education, Huzhou University, Huzhou, Zhejiang, China, 2 Postgraduate Program in Exercise and Sport Sciences, Khon Kaen University, Khon Kaen, Thailand

* 02148@zjhu.edu.cn

**Data Availability Statement:** Raw data of study has been uploaded as apart of supplementary information.

## Abstract

### Background

College students, especially during the COVID-19 pandemic, face substantial psychological stress. This study investigates the impact of Tai Chi (TC) practice on anxiety levels and theta oscillatory power activation characteristics among college students, aiming to provide empirical evidence for their psychological well-being.

### Methods

In this randomized controlled trial with 45 healthy college students, brainwave activity and changes in anxiety levels were measured. A 2 (TC group vs control group)×2 (pre-test vs post-test) factorial design was analyzed to explore TC's regulatory effects on brainwave activity and anxiety.

### Results

Following 12 weeks of TC practice, the TC group exhibited a significant decrease in state-trait anxiety differences (-6.14±14.33), state anxiety differences (-3.45±7.57), and trait anxiety differences (-2.68±7.43), contrasting with an increase in the control group. Moreover, contrasting with a decrease in the control group, TC group demonstrated significance increased theta oscillatory power in C3, C4, F4, P3, T7, and T8, and a significant negative correlations were observed between state anxiety and F4-θ (r = -0.31, p = 0.04), T7-θ (r = -0.43, p = 0.01), and T8-θ (r = -0.30, p = 0.05).

### Conclusion

The positive influence of TC on college students' psychological well-being and brain function is evident, leading to reduced anxiety levels and increased theta oscillatory activity. While encouraging further research to delve into the mechanisms of TC on anxiety and theta brainwave characteristics, the study recommends actively promoting TC practice among college students to enhance mental health and address post-pandemic psychological challenges.

**Funding:** This work was supported by the Zhejiang Province Philosophy and Social Sciences Leading Talent Cultivation Project (No. 23YJRC15ZD). The funders had no role in study design, data collection and analysis, decision to publish, or preparation of the manuscript.

**Competing interests:** The authors have declared that no competing interests exist.

## Introduction

Psychological challenges have intensified among college students, particularly in the face of escalated stress and anxiety stemming from the ongoing COVID-19 pandemic [1]. The intricate interplay between psychological well-being and the brain's response to stressors underscores the importance of comprehending the neural mechanisms at play [2]. In this context, anxiety, marked by aberrant neurotransmission within the brain, emerges as a central focus in both psychological and social research [2].

Theta (θ) brainwaves, characterized by a frequency of 4–8 hertz (Hz), represent a distinctive brainwave type associated with relaxation, meditation, and deep concentration, playing a pivotal role in cognitive functions such as emotion regulation, learning, and memory [3]. One study discovered that meditation practice increased theta activity in the prefrontal cortex and was inversely correlated with heightened anxiety levels [4, 5]. Conversely, another study explored the impact of psychological stress and anxiety on brain waves in college students and found that those with elevated anxiety levels exhibited increased theta activity [6]. Nevertheless, conflicting perspectives have been proposed, suggesting that the relationship between anxiety levels and brain electrical theta activity is intricate and may be influenced by individual differences, task characteristics, and measurement methods.

Furthermore, positive emotions have been associated with greater left-sided activation, while negative emotions are linked to greater right-sided activation [7]. An fMRI experiment indicated that individuals with generalized anxiety disorder may manifest functional abnormalities in the prefrontal and temporal lobes, particularly in the bilateral superior temporal gyrus and middle temporal gyrus, bilateral inferior frontal gyrus, left dorsolateral prefrontal lobe, bilateral subparietal lobule, bilateral anterior motor area, and overactivation in brain regions such as the anterior cingulate gyrus [6].

Tai Chi (TC), a traditional Chinese martial art renowned for its unique combination of movement, breath control, and emphasis on physical and mental equilibrium, has garnered widespread attention [8]. Despite extensive research on TC as a traditional mind-body exercise, there remains a significant gap in understanding its specific impact on the electroencephalogram (EEG) and anxiety levels of college students [9], especially within the challenging context of a pandemic. Consequently, this study aims to bridge this gap through a randomized controlled trial, shedding light on the potential mechanisms through which TC influences anxiety and theta brainwave characteristics among college students.

The primary hypothesis suggests that engaging in TC practice during the pandemic may have a dual effect: a reduction in anxiety scores and an augmentation of theta oscillatory activation. To rigorously test this hypothesis, the study utilizes a combination of EEG measurements and anxiety level assessments. The overarching goal is not only to discern differences between groups before and after TC practice but also to unravel the nuanced regulatory effects of TC on both brainwave activity and anxiety levels. Beyond these immediate objectives, the study endeavors to elucidate potential correlation mechanisms linking anxiety emotions with theta brainwaves, advancing our understanding of the intricate connections between mind, body, and the profound psychological implications of TC practice in the context of a global health crisis.

## Materials and methods

### Participants

For the sample size calculation, we used t-test by G*Power to ensure that an effect of 0.8 was obtained with $\alpha = 0.05$ and effect size of 0.85, requiring a minimum of 22 samples per group

[10]. The choice of test and effect size was based on our research question and prior knowledge in the field [11, 12]. Sixty college student volunteers were recruited from Huzhou University of China from March to April 2022, considering potential dropout rates. After baseline assessment, participants were stratified based on "gender," "age," and "BMI," with BMI categorized into four levels (below 18.5, 18.5–24.9, 25–29.9, and 30 kg/m$^2$) to avoid interference from BMI on anxiety [13] by stratified random sampling from the envelopes. Participants were then randomly assigned to the TC group or the control group by assistants, with allocation concealed from researchers.

In establishing the inclusion and exclusion criteria, careful considerations were made to ensure the study's validity. The inclusion criteria encompassed individuals aged between 18 and 22 years who were physically healthy, without regular medication, devoid of adverse lifestyle habits like smoking or excessive alcohol consumption, psychologically healthy with no symptoms of mental illness, and capable of participating in TC practice continuously for 12 weeks. These criteria were designed to create a homogeneous participant group, enhancing the internal validity of the study. Conversely, to isolate the effects of the intervention being studied and to avoid confounding variables, we did exclude participants who were currently using antidepressant medications or receiving formal psychological interventions, and participants who had certain pre-existing medical conditions that could potentially interfere with the study outcomes, such as severe neurological disorders or unstable chronic illnesses. We further excluded individuals who regularly engaged in professional exercises or with a reported history or risk of cardiovascular diseases or mental illness, and were unable to provide informed consent or who had language barriers that would prevent them from fully understanding the study procedures, contributing to the overall robustness and reliability of the study. By clearly defining and adhering to these exclusion criteria, the study population was made more homogeneous and the findings can be more accurately interpreted and potentially generalized to similar populations. All participants provided written informed consent, and ethical approval was obtained from the by the Center for Ethics in Human Research, Khon Kaen University (No. HE652012) and registration number ChiCTR2400081473. There were no deviations from the study protocol after approval was obtained. The research was conducted strictly in accordance with the approved plan, and no changes were made to the methods, procedures, or design throughout the study. Additional information regarding the ethical, cultural, and scientific considerations specific to inclusivity in global research is included in the Supporting Information.

## Exercise intervention

Before the formal exercise intervention, participants underwent a thorough assessment by completing the International Physical Activity Questionnaire (IPAQ) and Physical Activity Readiness Questionnaire (PARQ) twice, providing essential information on demographic data, medical history, and health behaviors.

The TC intervention, which was initiated in April 2022, comprised three weekly sessions over 12 consecutive weeks. To standardize the practice, two sessions of the 24-form simplified TC were instructor-led in the gym by a certified TC instructor at 4 pm on Tuesday and Thursday, and one session involved independent learning through videos at any free time, which was recorded for verification purposes. Each TC session had a duration of 45 minutes, encompassing a 5-minute warm-up, 35 minutes of practice, and a 5-minute relaxation period. Attendance was diligently recorded, with all analyzed participants achieving a 100% attendance rate. TC, recognized as a moderate-intensity aerobic exercise, incorporated elements such as controlled breathing and relaxation techniques. Proficiency growth was tracked, and participants maintained an average heart rate around 55% of their maximum during training [14]. In

contrast, the control group adhered to their regular daily activities without additional structured exercise during the intervention period. The specifics of these activities were not altered or restricted, allowing for a clear comparison between the TC intervention and the control group's routine activities.

## Data collection

**Anthropometry.** A standardized procedure was adopted for consistent and accurate anthropometric data in the Sports Anatomy and Physiology Laboratory of Huzhou University. Height was measured using a height and weight scale (Jiangsu Suhong Medical Equipment Co., Ltd., China), and weight was measured using the Huawei Smart Scale 3 Pro (Huawei Technologies Co., Ltd., China). Both measurements were taken twice by the same person and with the same instrument, with the average value recorded for each participant.

**Anxiety measurement.** The State-Trait Anxiety Inventory (STAI) was used to measure secondary outcome online, anxiety levels. The STAI consists of two subscales, with 40 descriptive questions in total. The first 20 items are from the State Anxiety Inventory (S-AI), half of which describe positive emotions, and the other half describe negative emotions. The cumulative scores range from 20 to 80, reflecting the degree of state or trait anxiety [15]. The STAI has high reliability in testing anxiety in college students (r = 0.85), with reliabilities of 0.89 for the State Anxiety Inventory and 0.94 for the Trait Anxiety Inventory [16]. In this study, the Cronbach α of the STAI-S was 0.950, and the Cronbach α of the STAI-T was 0.926 [17].

**Resting-state theta oscillatory power measurement.** Resting-state EEG serves as a widely accepted method for examining the brain's electrical activity during a passive state, offering insights into functional connections between different brain regions and identifying patterns associated with psychological states such as sleep, meditation, and cognitive processing. In this study, participants wore a 64-channel EEG cap in the Cognitive Neuroscience and Learning Sciences Laboratory of Huzhou University, recording data from 8 electrodes following the international 10–20 system (F3, F4, C3, C4, P3, P4, T7, and T8), to measure the primary outcome, with the ground electrode at the frontal center and the reference electrode at the nose tip [18]. To mitigate interference from posture changes and eye movements, participants maintained a still seated position with closed eyes for eight minutes. Data were recorded at a sampling rate of 1000Hz, and electrode impedance remained below 50kΩ [19]. Theta EEG changes associated with psychological states, particularly relaxation and meditation, were evaluated both at baseline and post-intervention. The Brain Vision Analyzer (version 2.2.0; Brain Products GmbH, Munich, Germany) facilitated data pre-processing, involving re-referencing to the average (eight selected electrodes) and band pass filtering (50Hz) to eliminate power line noise. Subsequently, IIR filters, encompassing a low-pass filter (cutoff frequency of 40Hz) and a high-pass filter (cutoff frequency of 0.5Hz), were applied to the data. Original filtering underwent semi-automatic verification per software design, and any remaining artifacts were manually identified and excluded. The dataset was divided into 2-second segments, excluding intervals with artifacts, and artifact-free segments were exported for microstate analysis [20]. Following the re-referencing of the signal to the average (eight selected electrodes), a fast Fourier transform (FFT) was conducted on the chosen data segments, calculating the power spectral density of theta oscillatory, and averaging the results for comprehensive analysis. These stringent data collection and processing steps ensure the reliability and validity of EEG measurements, aligning with the study's objectives to investigate the impact of TC practice on anxiety and brainwave activity during the COVID-19 pandemic.

The reliability and validity of EEG measures in assessing anxiety are integral to understanding the impact of TC practice on anxiety levels during the COVID-19 pandemic. High test-

retest reliability ensures the consistency of anxiety-related brainwave patterns over time, crucial for drawing meaningful conclusions about changes in neural activity. Validity, encompassing content and construct validity, ensures that EEG measures accurately capture anxiety states. The selected EEG electrodes based on the international 10–20 system and the use of established methodologies like Fast Fourier Transform enhance content and construct validity, respectively. These measures, including the assessment of theta oscillatory associated with relaxation and meditation, provide neurobiological correlates, offering an objective and comprehensive evaluation of anxiety. Such assessments contribute to understanding the mechanisms through which TC practice influences anxiety levels during the pandemic, providing valuable insights for holistic interventions.

## Statistical analysis

45 college students were randomly assigned to either the TC group or the control group via a sealed envelope within a 2 (TC group vs control group)×2 (pre-test vs post-test) factorial design, as shown Fig 1, and the reporting follows the general guidelines described in the Consolidated Standards of Reporting Trials (CONSORT) 2010 statement. All statistical analyses were conducted using IBM SPSS Statistics version 26.0 (IBM Corp., Armonk, NY). The normality of baseline data for the TC and control groups was assessed using the Shapiro-Wilk test. Data were presented as mean and standard deviation. An independent samples t-test, paired t-test and a two-way repeated ANOVA were used to determine significant differences between the two groups. Furthermore, canonical correlation and pairwise correlation were conducted to examine the correlation differences between anxiety levels and theta oscillatory power, with a significance level set at 0.05.

## Experimental results

### Participant characteristics

During the 12-week TC intervention (from April to June 2022), participants were categorized and stratified based on gender, age, and BMI, and then randomly assigned to the TC group or control group. Eight participants dropped out during the intervention, and an additional seven were excluded due to incomplete data collection. Finally, a total of 45 eligible participants remained, with 22 in the TC group and 23 in the control group. According to BMI standards, each group had one obese participant and two overweight participants; the TC group included 5 males, and the control group included 3 females. Age differences between the two groups were ignored as the mean values were very close. Moreover, there were no significant differences between the two groups in other baseline data (p>0.05), as shown in Table 1.

### Effects of 12 weeks of TC exercise on anxiety levels

After 12 weeks of TC exercise, the SAI scores in the TC group significantly decreased from 36.68±9.65 to 33.23±9.72 points (p* = 0.04). The control group's SAI scores increased from 38.48±9.38 to 42.74±13.23 points (p* = 0.03), TAI scores increased from 39.48±9.01 to 42.74 ±10.88 points (p* = 0.04), and STAI scores increased from 77.96±18.07 to 85.48±23.86 points (p* = 0.03). All these differences were significant.

In contrast, the control group showed an increase of 4.26±8.58 points. TAI scores in the TC group decreased from 38.86±10.06 to 36.18±10.75 points, while the control group increased by 3.26±7.07 points (p# = 0.03). Similarly, the STAI scores in the TC group decreased from 75.55 ±19.36 to 69.41±20.06 points, whereas the control group increased by 7.52±15.16 points (p# = 0.02). Significant differences were observed between the groups after intervention. Moreover,

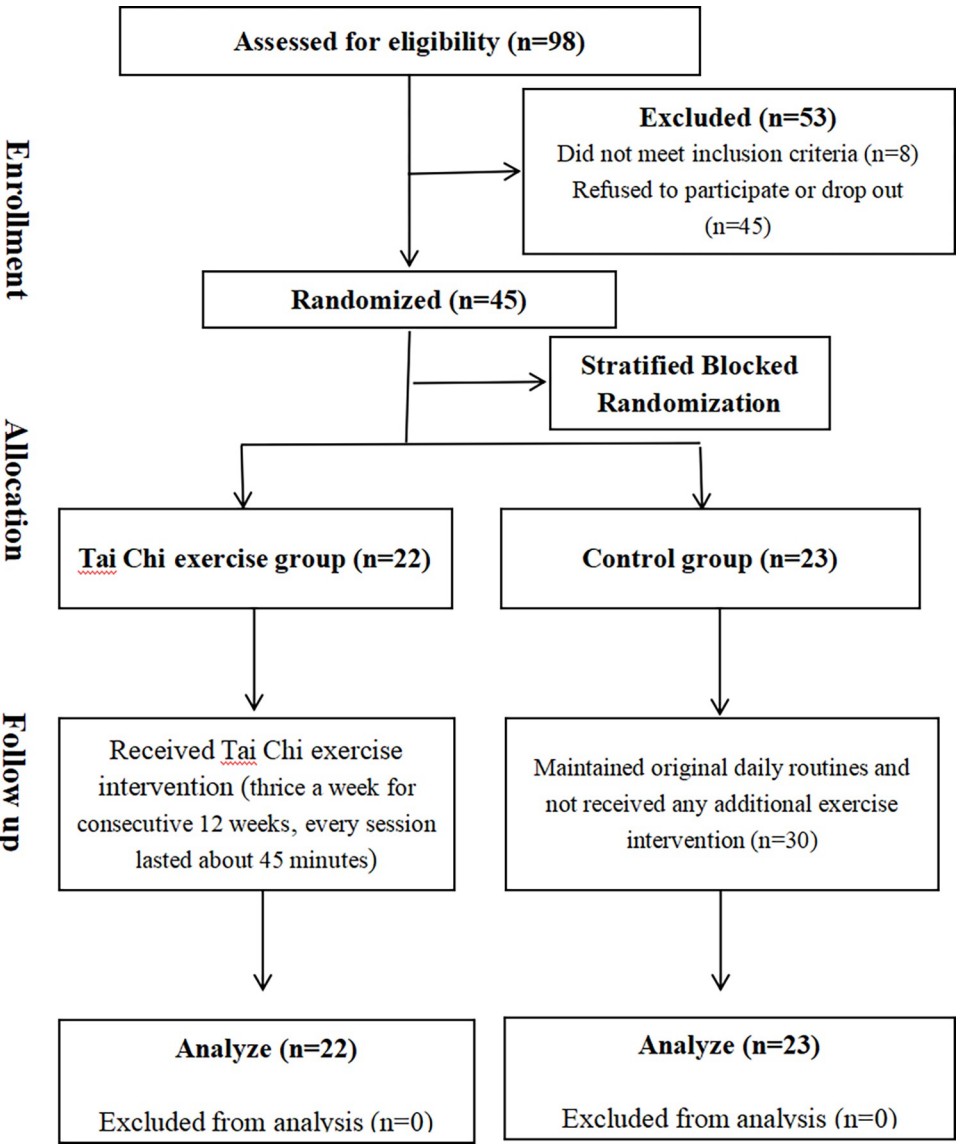

**Fig 1. Flowchart of the study protocol.**

there was an interaction effect of "Group and time" on the SAI (P$^{\dagger\ \dagger}$<0.01), TAI (P$^{\dagger}$ = 0.01) and STAI (P$^{\dagger\ \dagger}$<0.01) between the TC group and the control group after 12 weeks, as detailed in Table 2.

### Effects of 12 weeks of TC exercise on theta oscillatory power

According to the international 10–20 EEG system, C3 and C4 are located at the central sulcus on the left and right sides of the brain, F3 and F4 are positioned at the frontal lobes on the left and right sides, P3 and P4 are situated at the left and right parietal lobes, and T7 and T8 are positioned at the left and right temporal lobes.

After 12 weeks of TC exercise, the F4-θ oscillatory power in the TC group significantly increased from 0.48±0.31 to 0.91±0.97 ($10^{-2}$·μV·Hz) (p* = 0.02), the C4-θ oscillatory power of the control group decreased from 0.63±0.55 to 0.38±0.20 ($10^{-2}$·μV·Hz) (p* = 0.03), P3-θ oscillatory power decreased from 0.66±0.54 to 0.46±0.20 ($10^{-2}$·μV·Hz) (p* = 0.04), P4-θ oscillatory

**Table 1. Subject characteristics (n = 45).**

| Variables | Tai Chi group (n = 22) | Control group (n = 23) | P value |
|---|---|---|---|
| Age (years) | 20.09±0.29 | 20.70±0.56 | P<0.05 |
| Gender (male, female) | 7, 15 | 3, 20 | 0.14 |
| Height (cm) | 164.87±7.73 | 162.23±7.97 | 0.87 |
| Weight (kg) | 60.75±9.64 | 55.73±10.19 | 0.64 |
| S-AI (points) | 36.68±9.65 | 38.48±9.38 | 0.52 |
| T-AI (points) | 38.86±10.06 | 39.48±9.01 | 0.83 |
| STAI (points) | 75.55±19.36 | 77.96±18.07 | 0.55 |
| C3-θ ($10^{-2}$·μV·Hz) | 0.58±0.51 | 0.60±0.53 | 0.47 |
| C4-θ ($10^{-2}$·μV·Hz) | 0.60±0.53 | 0.63±0.55 | 0.41 |
| F3-θ ($10^{-2}$·μV·Hz) | 0.52±0.41 | 0.62±0.69 | 0.45 |
| F4-θ ($10^{-2}$·μV·Hz) | 0.48±0.31 | 0.57±0.68 | 0.80 |
| P3-θ ($10^{-2}$·μV·Hz) | 0.69±0.67 | 0.66±0.54 | 0.93 |
| P4-θ ($10^{-2}$·μV·Hz) | 0.72±0.63 | 0.71±0.57 | 0.91 |
| T7-θ ($10^{-2}$·μV·Hz) | 0.41±0.38 | 0.44±0.44 | 0.73 |
| T8-θ ($10^{-2}$·μV·Hz) | 0.37±0.31 | 0.44±0.38 | 0.35 |

Abbreviations: S-AI = State Anxiety Inventory, T-AI = Trait Anxiety Inventory, STAI = State Trait Anxiety Inventory.

power decreased from 0.71±0.57 to 0.42±0.17 ($10^{-2}$·μV·Hz) (p* = 0.01), T7-θ oscillatory power decreased from 0.44±0.44 to 0.26±0.14 ($10^{-2}$·μV·Hz) (p* = 0.02), and T8-θ oscillatory power decreased from 0.44±0.38 to 0.22±0.11 ($10^{-2}$·μV·Hz) (p** = 0.006), with all differences being statistically significant. When compared with the control group, differences in F3-θ (p# = 0.03) and F4-θ (p# = 0.02<0.05) oscillatory power were also significant. Moreover, there was an interaction effect of "Group and time" on theta oscillatory power in all positions (C3-θ, p† = 0.01; C4-θ, p† = 0.01; F3-θ, p† = 0.01; F4-θ, p†† <0.01; P3-θ, p† = 0.03; P4-θ, p† = 0.02; T3-θ, p† = 0.03; T4-θ, p† = 0.02) between the TC group and the control group after 12 weeks, as detailed in Table 3.

## Correlation between anxiety and theta oscillatory power

Furthermore, we not only observed the effects of 12 weeks of TC practice on theta oscillation power and anxiety, but also further analyzed the correlation between anxiety and theta

**Table 2. Effects of Tai Chi exercise on anxiety of college students.**

| Variables | Tai Chi group (n = 22) | | | Control group (n = 23) | | | Difference before and after (M±SD) | | Between-group comparisons | Group*Time | |
|---|---|---|---|---|---|---|---|---|---|---|---|
| | Pre (M±SD) | Post (M±SD) | P | Pre (M±SD) | Post (M±SD) | P | Tai Chi group | Control group | P | F | P |
| AI (points) | 36.68±9.65 | 33.23±9.72, | 0.04* | 38.48±9.38 | 42.74±13.23 | 0.03* | -3.45±7.57 | 4.26±8.58 | 0.01# | 10.20 | 0.00†† |
| T-AI (points) | 38.86±10.06 | 36.18±10.75 | 0.10 | 39.48±9.01 | 42.74±10.88 | 0.04* | -2.68±7.43 | 3.26±7.07 | 0.03# | 7.56 | 0.01† |
| STAI (points) | 75.55±19.36 | 69.41±20.06 | 0.06 | 77.96±18.07 | 85.48±23.86 | 0.03* | -6.14±14.33 | 7.52±15.16 | 0.02# | 9.62 | 0.00†† |

Abbreviations: S-AI = State Anxiety Inventory, T-AI = Trait Anxiety Inventory, STAI = State Trait Anxiety Inventory.

Note

* means significant within-group

# means significant between-group comparisons after the intervention

† means significant interaction effect of "Group and time".

**Table 3. Effects of Tai Chi exercise on theta oscillatory power of college students.**

| Variables $(10^{-2} \cdot \mu V \cdot Hz)$ | Tai Chi group (n = 22) | | | Control group (n = 23) | | | Difference before and after (M±SD) | | Between-group comparisons | Group*Time | |
|---|---|---|---|---|---|---|---|---|---|---|---|
| | Pre (M ±SD) | Post (M ±SD) | p | Pre (M ±SD) | Post (M ±SD) | p | Tai Chi group | Control group | P | F | P |
| C3-θ | 0.58±0.51 | 0.95±1.21 | 0.08 | 0.60±0.53 | 0.39±0.19 | 0.06 | 0.37±0.93 | -0.21±0.49 | 0.30 | 6.74 | 0.01† |
| C4-θ | 0.60±0.53 | 1.01±1.33 | 0.07 | 0.63±0.55 | 0.38±0.20 | 0.03* | 0.42±1.03 | -0.24±0.50 | 0.29 | 7.62 | 0.01† |
| F3-θ | 0.52±0.41 | 0.85±0.94 | 0.06 | 0.62±0.69 | 0.39±0.22 | 0.07 | 0.32±0.78 | -0.24±0.60 | 0.03# | 7.33 | 0.01† |
| F4-θ | 0.48±0.31 | 0.91±0.97 | 0.02* | 0.57±0.68 | 0.37±0.22 | 0.07 | 0.43±0.79 | -0.0±0.51 | 0.02# | 10.35 | 0.01†† |
| P3-θ | 0.69±0.67 | 1.19±1.72 | 0.12 | 0.66±0.54 | 0.46±0.20 | 0.04* | 0.50±1.44 | -0.20±0.46 | 0.27 | 4.87 | 0.03† |
| P4-θ | 0.72±0.63 | 1.14±1.54 | 0.13 | 0.71±0.57 | 0.42±0.17 | 0.01* | 0.42±1.28 | -0.30±0.51 | 0.38 | 6.17 | 0.02† |
| T7-θ | 0.41±0.38 | 0.71±1.03 | 0.15 | 0.44±0.44 | 0.26±0.14 | 0.02* | 0.30±0.95 | -0.18±0.35 | 0.11 | 5.13 | 0.03† |
| T8-θ | 0.37±0.31 | 0.72±1.15 | 0.14 | 0.44±0.38 | 0.22±0.11 | 0.01** | 0.35±1.08 | -0.22±0.32 | 0.08 | 5.69 | 0.02† |

Note

* means significant within-group

# means significant between-group comparisons after the intervention

† means significant interaction effect of "Group and time".

oscillation power. The analysis unveiled meaningful correlations between anxiety levels and theta oscillatory power. Notably, a significant negative correlations were observed between S-AI and F4-θ oscillatory power (r = -0.31, p = 0.04), S-AI and T7-θ oscillatory power (r = -0.43, p = 0.01), S-AI and T8-θ oscillatory power (r = -0.30, p = 0.05). Additionally, a significant negative correlation (r = -0.39, p = 0.01) emerged between STAI and T7-θ oscillatory power, as detailed in Table 4.

**Table 4. Correlation between HRV and cognitive performance variables (n = 45).**

| Variables | | △S-AI | △T-AI | △STAI |
|---|---|---|---|---|
| △C3-θ | Pearson Correlation | -0.24 | -0.10 | -0.19 |
| | p-value | 0.11 | 0.52 | 0.21 |
| △C4-θ | Pearson Correlation | -0.28 | -0.18 | -0.24 |
| | p-value | 0.06 | 0.25 | 0.11 |
| △F3-θ | Pearson Correlation | -0.18 | -0.04 | -0.11 |
| | p-value | 0.24 | 0.82 | 0.49 |
| △F4-θ | Pearson Correlation | -0.31* | -0.20 | -0.27 |
| | p-value | 0.04 | 0.20 | 0.07 |
| △P3-θ | Pearson Correlation | -0.16 | -0.07 | -0.13 |
| | p-value | 0.30 | 0.66 | 0.41 |
| △P4-θ | Pearson Correlation | -0.12 | -0.02 | -0.08 |
| | p-value | 0.43 | 0.89 | 0.59 |
| △T7-θ | Pearson Correlation | -0.43** | -0.28 | -0.39** |
| | p-value | 0.01 | 0.06 | 0.01 |
| △T8-θ | Pearson Correlation | -0.30* | -0.19 | -0.27 |
| | p-value | 0.05 | 0.21 | 0.08 |

Abbreviations: S-AI = State Anxiety Inventory, T-AI = Trait Anxiety Inventory, STAI = State Trait Anxiety Inventory.

Note: △ means the difference between pre-test and post-test; use * for p<0.05 and ** for p<0.01.

## Discussion

The study sought to assess the effects of a 12-week TC exercise intervention on anxiety and theta oscillatory in college students amidst the challenges posed by the COVID-19 pandemic. The results demonstrated that, in comparison to the control group, participation in TC exercises correlated with a reduction in anxiety levels and a significant increase in theta oscillatory activity across various brain regions, including C3, C4, F4, P3, T7, and T8. Several studies have shown that alterations in theta brainwave activity can be associated with anxiety disorders. In healthy individuals, theta oscillations play important roles in memory formation, emotional processing, and attention. However, in those with anxiety disorders, there appears to be a dysregulation of theta activity. For example, some research has found that individuals with generalized anxiety disorder exhibit increased theta power in certain brain regions, such as the prefrontal cortex and hippocampus. This increased theta activity may be related to heightened emotional reactivity and rumination, which are common features of anxiety [21, 22]. Another line of research has investigated the role of theta oscillations in the amygdala, a key brain region involved in fear and anxiety. Studies have shown that abnormal theta activity in the amygdala may contribute to the development and maintenance of anxiety disorders [23]. Furthermore, some studies have used neurofeedback techniques to train individuals to modulate their theta brainwaves. Preliminary results suggest that by learning to control theta activity, it may be possible to reduce anxiety symptoms [24].

Psychological anxiety, defined by inner unrest, worry, and tension, constituted the primary focus of our inquiry. TC, classified as a mild to moderate aerobic exercise, combines physical and mental training through the integration of stress increment, lower limb fitness promotion, dynamic weight transfer, relaxed breathing, focused attention, and meditation techniques. The study's outcomes suggest the efficacy of TC in alleviating anxiety, aligning with previous research indicating positive effects on mental health. For instance, a 12-week 24-form TC significantly ameliorated depression and anxiety in non-clinical college students [25]. Similarly, a meta-analysis demonstrated significant improvement in anxiety symptoms among drug addicts practicing TC or Qigong compared to control groups [26]. Another meta-analysis, encompassing 15 randomized controlled trials, revealed that meditative movements in TC, Qigong, and yoga could reduce the severity of depression and anxiety [27]. Additionally, a study found that the 42-form Yang-style TC significantly reduced depression scores in elderly individuals and enhanced parasympathetic nervous system activity [28]. However, a single-arm experiment reported no significant changes in perceived stress and anxiety after 16 weeks of TC courses among college students, potentially attributable to the exercise's low intensity or limitations of the Generalized Anxiety Disorder-7 scale [29]. Notably, the COVID-19 pandemic has substantially impacted physical activity and neuro-psychological aspects, exacerbating challenges in mental health [30].

Delving into the theoretical underpinnings of the relationship between TC, anxiety, and brainwave activity, our study aimed to enhance the understanding of how TC influences anxiety levels, and highlighted the relatively weak correlation (r: 0.3–0.4) between anxiety levels and theta oscillatory power. Firstly, while there was an observable connection between anxiety and theta oscillatory power, it was not as strong as might have been expected. This suggests that while theta oscillations may play a role in anxiety, other factors were likely also at play. It implied that anxiety was a complex phenomenon that cannot be solely attributed to changes in theta power. It also indicated that more comprehensive studies are required better to understand the relationship between brainwave activity and anxiety. Moreover, this finding should be considered when interpreting the results of interventions aimed at reducing anxiety through modulating theta oscillations. The relatively weak correlation suggests that such

interventions may not significantly impact anxiety solely through changes in theta power. However, it does not rule out the possibility that they could still have some beneficial effects in combination with other approaches.

The identification of increased theta oscillatory activation following TC practice suggests a neuro-biological basis for the observed anxiety-alleviating effects, aligning with theories proposing the role of theta oscillatory in emotion regulation and stress reduction. Despite indications that anxiety disorder patients may exhibit increased theta oscillatory in specific situations due to a focus on negative thoughts and emotions, difficulty shifting attention, heightened negative emotions, and sensitivity to stress, theta oscillatory generally manifest during relaxation, meditation, and light sleep within the frequency range of 4 to 8 Hz. This state is typically associated with inner calmness, relaxation, and focused attention, traits often targeted by meditation and relaxation techniques. As a mind-body exercise, TC integrates physical, cognitive, and meditative elements, emphasizing conscious control of body movement through postures and deep breathing [31].

A previous TC study has observed an increase in theta and delta oscillatory during breath retention and slow breathing, indicating activation of parasympathetic nervous system activity [32]. In a single-arm experiment with 38 adults, a 20-minute TC session tended to reduce anxiety and increase theta oscillatory activity, as assessed by the STAI, electrocardiogram, and electroencephalogram [33]. Our study further substantiates these findings by demonstrating decreased anxiety levels and increased theta oscillatory power in the TC group, while the control group exhibited a trend of increasing anxiety levels and decreasing theta oscillatory power. This correlation strengthens the reliability of theta oscillatory as an indicator of anxiety. It's crucial to acknowledge that the relationship between theta oscillatory and psychological states is complex, with potential individual differences across different populations and contexts. Nevertheless, our study supports the positive impact of TC on anxiety and the theta oscillatory characteristics of the brain.

The meditative component of TC contributes to beneficial to specific brain structures, with studies indicating improved inter-hemispheric metabolism, neural integration, and memory enhancement, particularly in the frontal lobe [34]. Our study found a significant increase in theta oscillatory power in the right frontal lobe after 12 weeks of TC practice, and significant negative correlations between state anxiety and F4-θ oscillatory power, aligning with previous literature suggesting that TC increases theta oscillatory power [33]. According to the criterion that "asymmetric activation of the frontal area can distinguish emotional states," positive emotions are associated with greater activation in the left brain area, while negative emotions are associated with greater activation in the right brain area. Our results support this criterion, indicating that TC practice increased theta oscillatory activation in the right frontal lobe, promoting relaxation and meditation, and contributing to anxiety relief. Electrophysiological studies also highlight that different emotions are associated with distinct brainwave patterns in the frontal lobe region [35].

Furthermore, the TC group differed significantly from the control group in theta oscillatory power in the central sulcus, right frontal lobe, left parietal lobe, and bilateral temporal lobes. This aligns with a clinical fMRI experiment showing that patients with generalized anxiety disorder may exhibit functional abnormalities in the frontal lobe and temporal lobe, primarily manifesting as overactivation in various brain regions such as the bilateral temporal gyrus, bilateral frontal subcallosal gyrus, left anterior cingulate gyrus, bilateral postcentral gyrus, bilateral precentral gyrus, and anterior part of the cingulate gyrus [36]. TC, considered a positive mindfulness meditation practice, exhibited increased theta oscillatory to counteract anxiety, similar to increased theta oscillatory in mindfulness in the parietal lobe [37].

TC also exerts a shaping effect on brain structure. Previous research indicated that TC practitioners improved inter-hemispheric metabolism and neural integration, enhancing the network connection between the frontal cortex and central motor cortex, thus improving brain function [36]. Functional magnetic resonance imaging (fMRI) studies suggest that TC practitioners enhance memory through structural and functional changes in the frontal lobe. For instance, elderly TC practitioners exhibited enhanced activity in the left frontal middle gyrus related to working memory during resting states [38]. Structural changes were observed in the cortical thickness of the parietal cortex and occipital cortex [39]. In a study with college students, a significant increase in the cortical thickness of the left upper parietal lobe's frontal cortex was found after 8 weeks of TC practice. Additionally, a comparison of gray matter volumes before and after different exercises revealed a significant increase in gray matter volume in various brain regions in the TC group [40]. Notably, while electrodes are placed in pairs, the degree of change in theta oscillatory power may vary between different electrodes (such as C3 and C4, F3 and F4), emphasizing the intricate and interconnected nature of emotion generation networks [41]. Furthermore, while TC is generally regarded as a safe practice, there exists a minor risk of injury. For example, participants may encounter muscle strains or joint discomfort if they execute the movements improperly or exert themselves excessively. Moreover, some individuals might perceive the time commitment associated with regular TC practice as burdensome, potentially leading to stress or dissatisfaction [42].

Employing a randomized controlled trial design, our study investigated the impact of a 12-week TC practice on the mental health of college students during the COVID-19 pandemic, concurrently collecting physiological data on brain theta oscillatory. While the research contributes valuable insights into the interplay between TC, anxiety, and brain structures, several limitations must be acknowledged. First, a small sample size can limit the generalizability of our findings. Our results may not be representative of the broader population from which our sample was drawn, or they may not hold true in different settings or for different subgroups. Another concern with a small sample size is the increased risk of sampling bias. Despite these limitations, we took several steps to mitigate the impact of the small sample size. We used rigorous study design and statistical methods to ensure that our analyses were as accurate and reliable as possible. We also carefully considered the potential sources of bias and took steps to minimize them. Future research with larger sample sizes would be valuable to confirm and extend our findings. Secondly, another significant challenge was the pandemic. Students experienced disruptions to their academic routines, transitioned to online learning, and had to adapt to new pedagogical methods and technologies. The uncertainty surrounding the duration and effectiveness of online education also contributed to heightened anxiety and stress; lockdowns, campus closures, and social distancing measures resulted in social isolation, increased feelings of loneliness, depression, and anxiety; furthermore, widespread economic pressures coupled with uncertainties regarding future employment prospects during the pandemic exacerbated feelings of anxiety and despair. These challenges directly impact students' mental health, which subsequently influences their participation in trials. Nevertheless, despite these adversities, many students exhibited resilience and adaptability by proactively seeking support from mental health resources while reinforcing their motivation to engage in trials. To address this issue further studies could consider implementing a psychological comfort group for additional validation. Moreover, the 12-week study duration constrains insights into the long-term psychological effects of TC on college students. Despite these limitations, this research represents a pioneering effort as it is the first to employ TC as an intervention for mental health concerns among healthy college students during a pandemic. The findings highlight positive effects of TC on reducing anxiety levels and enhancing cognitive efficacy among

college students throughout the COVID-19 pandemic—an area that warrants further investigation and promotion within this demographic.

## Conclusions

This study delved into the impact of TC exercise on anxiety and theta oscillatory characteristics in college students during the pandemic. Through a 12-week randomized controlled trial, we discerned positive effects on the mental health and brain function of college students engaging in TC exercise. Notably, anxiety levels significantly decreased after the 12-week TC program, accompanied by a noteworthy increase in theta oscillatory power, indicative of relaxation, meditation, and emotional balance. While these findings highlight the influence of TC on brain activity patterns, exploring the long-term effects on anxiety and theta oscillatory characteristics, along with unraveling the underlying mechanisms, warrants further investigation. As a practical recommendation, we advocate for the active promotion of TC exercise among college students to fortify their mental health and assist them in navigating the psychological challenges of the post-pandemic era.

## Supporting information

**S1 File. Clinical trial protocol.**
(PDF)

**S2 File. Raw data of study.**
(PDF)

**S3 File. Inclusivity in global research questionnaire.**
(PDF)

## Acknowledgments

We would like to thank all research assistants who helped us conduct this study and all volunteers that participated in it.

## Author Contributions

**Conceptualization:** Min Wang, Shuxun Chi, Xingze Wang, Tongling Wang.

**Data curation:** Min Wang, Xingze Wang.

**Formal analysis:** Min Wang, Shuxun Chi.

**Funding acquisition:** Xingze Wang.

**Investigation:** Min Wang.

**Methodology:** Min Wang, Shuxun Chi, Xingze Wang, Tongling Wang.

**Resources:** Min Wang.

**Software:** Min Wang.

**Supervision:** Xingze Wang.

**Validation:** Min Wang.

**Visualization:** Min Wang, Xingze Wang, Tongling Wang.

**Writing – original draft:** Min Wang.

**Writing – review & editing:** Min Wang, Shuxun Chi, Xingze Wang.

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
