## [Decision Letter · Decision Letter 0]

27 Aug 2024

PONE-D-24-02481Effects of Tai Chi on anxiety and theta oscillation power in college students during the COVID-19 pandemic: a randomized controlled trialPLOS ONE

Dear Dr. WANG,

Thank you for submitting your manuscript to PLOS ONE. After careful consideration, we feel that it has merit but does not fully meet PLOS ONE’s publication criteria as it currently stands. Therefore, we invite you to submit a revised version of the manuscript that addresses the points raised during the review process.

The reviewers have provided valuable feedback that will help strengthen your work. Both Reviewer1 and 2 have raised important points regarding the methodology, statistical analysis, and clarity of your manuscript. I agree with Reviewer 1 in providing further clarification on certain aspects of the study design, including sample size calculation, stratification, and the presentation of results. Additionally, Rev1 gaved suggestions for improving the tables and considering adjustments for multiple outcomes. Reviewer 2 has provided constructive feedback on the relationship between the COVID-19 pandemic and your trial, the connection between theta brainwaves and anxiety, and the need for a more detailed description of your Tai Chi practice. I think your manuscript would benefit from the expansion of the discussion around potential confounding factors and ensuring the exclusion criteria are clearly defined.

We encourage you to carefully address these comments in your revised manuscript to enhance the clarity, rigor, and overall impact of your study. 

We look forward to receiving your revised manuscript.

Kind regards,

Valentina Bruno

Academic Editor

PLOS ONE

Journal requirements: 1. When submitting your revision, we need you to address these additional requirements. Please ensure that your manuscript meets PLOS ONE's style requirements, including those for file naming. The PLOS ONE style templates can be found at https://journals.plos.org/plosone/s/file?id=wjVg/PLOSOne_formatting_sample_main_body.pdf and https://journals.plos.org/plosone/s/file?id=ba62/PLOSOne_formatting_sample_title_authors_affiliations.pdf. 2. Please include a complete copy of PLOS’ questionnaire on inclusivity in global research in your revised manuscript. Our policy for research in this area aims to improve transparency in the reporting of research performed outside of researchers’ own country or community. The policy applies to researchers who have travelled to a different country to conduct research, research with Indigenous populations or their lands, and research on cultural artefacts. The questionnaire can also be requested at the journal’s discretion for any other submissions, even if these conditions are not met.  Please find more information on the policy and a link to download a blank copy of the questionnaire here: https://journals.plos.org/plosone/s/best-practices-in-research-reporting. Please upload a completed version of your questionnaire as Supporting Information when you resubmit your manuscript. 3. Thank you for stating the following financial disclosure:  [This work was supported by the Zhejiang Province Philosophy and Social Sciences Leading TalentCultivation Project (No. 23YJRC15ZD).].  Please state what role the funders took in the study.  If the funders had no role, please state: ""The funders had no role in study design, data collection and analysis, decision to publish, or preparation of the manuscript."" If this statement is not correct you must amend it as needed. Please include this amended Role of Funder statement in your cover letter; we will change the online submission form on your behalf. 4. Thank you for stating the following in the Acknowledgments Section of your manuscript: [We would like to thank all research assistants who helped us conduct this study and all volunteers that participated in it, and the publication of this article: the Major Cultivation Project of Leading Talents in Philosophy and Social Sciences in Zhejiang Province (Project Number: 23YJRC15ZD), China.]We note that you have provided funding information that is not currently declared in your Funding Statement. However, funding information should not appear in the Acknowledgments section or other areas of your manuscript. We will only publish funding information present in the Funding Statement section of the online submission form. Please remove any funding-related text from the manuscript and let us know how you would like to update your Funding Statement. Currently, your Funding Statement reads as follows:  [This work was supported by the Zhejiang Province Philosophy and Social Sciences Leading TalentCultivation Project (No. 23YJRC15ZD).].   Please include your amended statements within your cover letter; we will change the online submission form on your behalf. 5. In the online submission form, you indicated that [If someone need raw data such as EEG, can contact the first author of this article by email.]. All PLOS journals now require all data underlying the findings described in their manuscript to be freely available to other researchers, either 1. In a public repository, 2. Within the manuscript itself, or 3. Uploaded as supplementary information.This policy applies to all data except where public deposition would breach compliance with the protocol approved by your research ethics board. If your data cannot be made publicly available for ethical or legal reasons (e.g., public availability would compromise patient privacy), please explain your reasons on resubmission and your exemption request will be escalated for approval.  6. Please amend your list of authors on the manuscript to ensure that each author is linked to an affiliation. Authors’ affiliations should reflect the institution where the work was done (if authors moved subsequently, you can also list the new affiliation stating “current affiliation:….” as necessary). 7. Please include captions for your Supporting Information files at the end of your manuscript, and update any in-text citations to match accordingly. Please see our Supporting Information guidelines for more information: http://journals.plos.org/plosone/s/supporting-information. 

Reviewers' comments:

Reviewer's Responses to Questions

**Comments to the Author**

1. Is the manuscript technically sound, and do the data support the conclusions?

Reviewer #1: Partly

Reviewer #2: Partly

Reviewer #3: Yes

2. Has the statistical analysis been performed appropriately and rigorously? 

Reviewer #1: No

Reviewer #2: N/A

Reviewer #3: Yes

3. Have the authors made all data underlying the findings in their manuscript fully available?

Reviewer #1: Yes

Reviewer #2: Yes

Reviewer #3: Yes

4. Is the manuscript presented in an intelligible fashion and written in standard English?

Reviewer #1: Yes

Reviewer #2: Yes

Reviewer #3: Yes

5. Review Comments to the Author

Reviewer #1: What test and what effect size were used for sample size calculation?

For such a small sample size, it is almost impossible to stratify the random sample over sex, age and BMI groups. The authors did not mention how many age groups stratified. Assume there are 3 age groups, there will be 2x3x5=30 strata to divide the total N=60.

If the sample was stratified by age, why is there significant difference between the two treatment arms as shown in Table 1? Was age adjusted in the primary analysis?

Tables 2 and 3, better add pre-post change for each group. In addition, better add p values for within-group and between-group comparisons. The F values and p values associated with the F tests are not friendly to general audience and can be omitted.

There are two many outcomes for a limited sample size. The p values better be adjusted for multiple outcomes and correlations.

Minor concerns:

Specify the decimal numbers for all p values <0.05.

Line 85 There are only 4 BMI groups instead 5 groups.

Table 1 omit the column for 95% CI.

Limitation of the sample size should be discussed.

Reviewer #2: - The relationship between the COVID-19 pandemic and the trial requires further elaboration. Specifically, was the pandemic ongoing during the study? If so, how did you measure its impact on the participants and the outcomes? Furthermore, how were the participants affected by the pandemic, both psychologically and behaviorally?

- The connection between theta brainwaves and anxiety needs to be supported by additional clinical studies. Please provide a more detailed discussion of the existing literature on this relationship, offering insights into how theta oscillations are implicated in anxiety disorders.

- It is essential to hypothesize both potential positive and negative outcomes before initiating a clinical trial. In the current version, the introduction section primarily focuses on the probable positive effects of Tai Chi. Please ensure that the possibility of neutral or negative effects is also acknowledged.

- The description of Tai Chi practice should be more detailed. Information regarding the timing of the sessions (e.g., morning vs. evening) and the standardization of the practice should be included. Furthermore, clarify how data was recorded, including any specific protocols or technologies used.

- The exclusion criteria need to be presented in more detail. For instance, did you exclude participants who were using antidepressant medications or receiving psychological interventions? Clearly specifying such criteria is crucial for understanding the generalizability of the findings.

- If the Chinese version of the STAI was used, please provide information regarding its reliability, such as the Cronbach's alpha coefficient and any validation data for the scale within the study population.

- According to your analysis, the correlation between anxiety levels and theta oscillatory power was relatively weak (with correlation coefficients ranging from 0.3 to 0.4). I recommend emphasizing this finding in the results section, including a more thorough interpretation of its significance.

- In the discussion, you mention "college students amidst the challenges posed by the COVID-19 pandemic." Please describe in greater detail what specific challenges the students faced during the pandemic and how these may have impacted their mental health and engagement with the trial.

- Please clarify how other potential confounding factors that could influence theta brainwaves were controlled or excluded in your analysis. Ensuring that these factors are adequately accounted for is vital to the study's internal validity.

Reviewer #3: Dea Author

This study investigates the effects of a 12-week Tai Chi (TC) practice on the mental health of university students during the COVID-19 pandemic and presents significant data and findings. The research evaluates the impact of TC exercise on anxiety levels and brain theta oscillatory characteristics. Conducted as a randomized controlled trial, it also provides physiological data related to brain theta oscillation.

The article has been meticulously planned, implemented, and written. The findings and analyses presented are poised to make a significant contribution to the existing literature.

I believe this study will provide valuable insights in its field. In my opinion, the study is suitable for publication in its current form.

best regards

6. PLOS authors have the option to publish the peer review history of their article (what does this mean?). If published, this will include your full peer review and any attached files.

Reviewer #1: No

Reviewer #2: No

Reviewer #3: No

---

## [Author Response · Author response to Decision Letter 0]

13 Sep 2024

Dear editor and reviewers, thank you very much for the modification suggestions given by experts. We have made modifications according to the comments. For details, please refer to the letter of "Response to Reviewers" and “Revised Manuscript with Track Changes”.

---

## [Decision Letter · Decision Letter 1]

15 Oct 2024

Effects of Tai Chi on anxiety and theta oscillation power in college students during the COVID-19 pandemic: a randomized controlled trial

PONE-D-24-02481R1

Dear Dr. WANG,

We’re pleased to inform you that your manuscript has been judged scientifically suitable for publication and will be formally accepted for publication once it meets all outstanding technical requirements.

Kind regards,

Valentina Bruno

Academic Editor

PLOS ONE

Additional Editor Comments (optional):

Reviewers' comments:

Reviewer's Responses to Questions

**Comments to the Author**

1. If the authors have adequately addressed your comments raised in a previous round of review and you feel that this manuscript is now acceptable for publication, you may indicate that here to bypass the “Comments to the Author” section, enter your conflict of interest statement in the “Confidential to Editor” section, and submit your "Accept" recommendation.

Reviewer #1: All comments have been addressed

Reviewer #2: All comments have been addressed

2. Is the manuscript technically sound, and do the data support the conclusions?

Reviewer #1: (No Response)

Reviewer #2: Yes

3. Has the statistical analysis been performed appropriately and rigorously? 

Reviewer #1: (No Response)

Reviewer #2: Yes

4. Have the authors made all data underlying the findings in their manuscript fully available?

Reviewer #1: (No Response)

Reviewer #2: Yes

5. Is the manuscript presented in an intelligible fashion and written in standard English?

Reviewer #1: (No Response)

Reviewer #2: Yes

6. Review Comments to the Author

Reviewer #1: All my concerns are addressed.

Reviewer #2: Dear Authors,

Thank you for your response to my suggestions. I have reviewed the revised version of your manuscript and would like to express my appreciation for the thorough changes you have made. The revisions have addressed the concerns and suggestions provided, and I am pleased with the updated content.

7. PLOS authors have the option to publish the peer review history of their article (what does this mean?). If published, this will include your full peer review and any attached files.

Reviewer #1: No

Reviewer #2: **Yes: **Yasemin Cayir

---

## [Editor Report · Acceptance letter]

23 Oct 2024

PONE-D-24-02481R1 

PLOS ONE

Dear Dr. Wang, 

I'm pleased to inform you that your manuscript has been deemed suitable for publication in PLOS ONE. Congratulations! Your manuscript is now being handed over to our production team.

Kind regards, 

on behalf of

Dr. Valentina Bruno 

Academic Editor

PLOS ONE